# Depressive and Anxiety Symptoms among Children and Adolescents in Rural China: A Large-Scale Epidemiological Study

**DOI:** 10.3390/ijerph19095026

**Published:** 2022-04-20

**Authors:** Qi Jiang, Xinshu She, Sarah-Eve Dill, Sean Sylvia, Manpreet Kaur Singh, Huan Wang, Matthew Boswell, Scott Rozelle

**Affiliations:** 1Freeman Spogli Institute for International Studies, Stanford University, Stanford, CA 94305, USA; qi_jiang@berkeley.edu (Q.J.); sedill@stanford.edu (S.-E.D.); huanw@stanford.edu (H.W.); kefka@stanford.edu (M.B.); rozelle@stanford.edu (S.R.); 2School of Medicine, Stanford University, Stanford, CA 94305, USA; xinshe@stanford.edu; 3Gillings School of Global Public Health, University of North Carolina at Chapel Hill, Chapel Hill, NC 27514, USA; sean.sylvia@unc.edu; 4Stanford Pediatric Mood Disorders Program, Stanford University, Stanford, CA 94305, USA

**Keywords:** mental health, depression, anxiety, learning anxiety, gender, age, socioeconomic status, school-age children, rural China, epidemiological study

## Abstract

Although children living in low- and middle-income countries (LMICs) account for 90% of the global population of children, depression, and anxiety among children in LMICs have been understudied. This study examines the prevalence of depression and anxiety and their associations with biological and psychosocial factors among children across China, with a focus on rural areas. We conducted a large-scale epidemiological study of depression and anxiety among 53,421 elementary and junior high school-aged children across China. The results show that 20% are at risk for depression, 6% are at risk for generalized anxiety, and 68% are at risk for at least one type of anxiety. Girls and junior high school students show a higher risk for both depression and anxiety symptoms, while socioeconomic status has varying associations to depression and anxiety symptoms. Our results also show consistent correlations between depression and anxiety symptoms and standard math test scores. These findings underscore the importance of identification, prevention, and treatment of youth depression and anxiety in underdeveloped areas. As China constitutes 15% of the global population of children under age 18, this study offers valuable information to the field of global mental health.

## 1. Introduction

A growing body of research has highlighted the importance of child mental health for optimal social and academic functioning. International studies show that mental health problems tend to appear early among elementary and junior high school students and can impede self-confidence, social interactions, and cognitive ability [1]. As a result, mental health problems, such as anxiety and depression, are well known for negatively affecting learning and academic achievement [1]. Moreover, since depression and anxiety problems may affect academic achievement, and may persist beyond childhood or adolescence and into adulthood, recent studies suggest an association between childhood-onset mental health problems and economic and social outcomes in later life [2].

The global prevalence of depression and anxiety symptoms is high among children and adolescents [3]. A meta-analysis of 41 studies of children aged 4–18 years in 27 countries around the world found a global prevalence of 13.4% for any mental disorder, 65% for anxiety, and 26% for depression [4]. Notably, the prevalence of depression and anxiety problems among children and adolescents varies widely between countries, from 18.1% in India to 39.4% in Mexico [3]. Much of this variation is thought to be due to differences in local context, which both creates specific sources of distress and impairment and affects how symptoms are interpreted and diagnosed [5].

China, and particularly rural China, presents a unique context in which to examine childhood onset of depression and anxiety symptoms due to both long-standing social norms and the vast economic and social changes that have occurred in the last four decades. There are three factors in particular that may uniquely shape the mental health of school-aged children in rural China. The first factor is gender. Past international research has found higher incidence of depression and anxiety problems among girls compared to boys [6,7,8]. In China, discrimination against girls may create a larger gender gap, especially in rural areas. Rural China has a long-lasting tradition of male preference, exhibited though behaviors such as sex-selective abortion and neglect for the healthcare needs of girls [9,10]. Studies have also found that caregivers in rural areas of China tend to invest more resources in male children while neglecting support for girls [10].

A second factor is the stressful nature of China’s educational system. The “exam culture” of education in China has existed since the 7th century, and it continues through state systems of high-stakes exams, such as the ultra-competitive college entrance exam and the equally rigorous high school entrance exam [11]. The role of exams in academic attainment increases academic pressure on students in China, which may exacerbate pre-existing vulnerabilities to depression and anxiety. This pressure may be greater for students in junior high school compared to elementary school students, as access to high school education is not guaranteed by China’s nine-year compulsory education system. Additionally, the pressure may be even greater for junior high schools students in rural areas, where fewer available slots in academic high schools lead to greater competition for entry [12].

Finally, rural-to-urban migration in China complicates the picture of child and adolescent mental health. In China’s rapid development and urbanization, a large share of rural parents have migrated from their homes to urban areas in search of higher wages [13]. Due to China’s household registration (hukou) system, however, in most cases it is impossible for the family to move together into the city, since rural children usually cannot access urban social services, including education and healthcare [14]. As a result, many rural children are “left behind” in rural areas under the care of surrogate caregivers, typically paternal grandparents [15].

The literature points to two competing effects of parental migration on the mental health of rural left-behind children: increased household income and decreased parental care. On the one hand, the increased income from migrant work can raise the socioeconomic status (SES) of family members left behind in rural areas. Higher SES, in turn, has been shown to be a protective factor for depression and anxiety symptoms, since higher SES usually corresponds to a better family environment and more access to mental healthcare [16]. On the other hand, decreased parental care due to out-migration may worsen the mental health of left-behind children, since, among other things, they would not have access to face-to-face emotional support from their parents [17]. Similarly, although past studies have found higher parental education (another key aspect of SES) to be a protective factor for child mental health, [16] this protective effect may be offset by the fact that in rural China, parents with higher education levels are relatively more likely to out-migrate [18]. Therefore, it is possible that the positive effects of higher SES on child and adolescent depression and anxiety symptoms is offset in rural China by reduced parental care due to out-migration.

Despite the large number of studies on child depression and anxiety internationally, evidence of how factors unique to China (especially rural areas of China) shape depression and anxiety symptoms among children and adolescents is limited and mixed. For example, some studies have found that that girls are more likely to have anxiety and depression than boys, [17] whereas other studies have found no gender differences in either anxiety or depression [19,20]. This may be due to the fact that past studies have mainly relied on small samples, which can undermine the reliability of the results. Additionally, previous studies of depression and anxiety by age and socioeconomic status (SES) have been limited to specific subgroups such as left-behind children, or to broad comparisons between rural and urban children [21].

The overall goal of this paper is to answer three major questions about depression and anxiety symptoms among children and adolescents across rural China. First, what is the overall prevalence of risk for depression and anxiety among children and adolescents across China? Second, how does risk for depression and anxiety, as well as the severity of math anxiety, vary by gender, grade level, and SES? Third, after controlling for demographic characteristics, what is the correlation between depression and anxiety symptoms and student academic performance? We hypothesize that the prevalence of depression and anxiety symptoms will be higher than the global average, and that girls, children in higher grade levels, and children from families with lower SES will show higher prevalence of symptoms for both depression and anxiety. We also hypothesize that symptoms of depression and anxiety will be correlated with lower levels of standard math test scores.

## 2. Methods

### 2.1. Procedures and Data

The data used in this study were aggregated from seven datasets across China. In our study, datasets that measure the same mental health outcome (depression, anxiety, or math anxiety) were combined. Specifically, we use Dataset 1 to examine depression, pooled data from Datasets 2–5 to examine anxiety, and pooled data from Dataset 6 and 7 to examine math anxiety. The distribution of students and schools in all seven datasets, as well as previous publications using each dataset, are presented in Table 1.

Dataset 1 comes from the second survey wave of the China Family Panel Study (CFPS). The CFPS is a nationally representative, household-based social survey conducted by the Institute of Social Science Surveys (ISSS) at Peking University. The second survey wave of CFPS, which was conducted in 2012, surveyed respondents in 25 of China’s 31 provinces (excluding Xinjiang, Tibet, Qinghai, Inner Mongolia, Ningxia, and Hainan), a sampling frame that represents 95% of China’s population.

The remaining six datasets (Datasets 2–7) come from school-based surveys conducted in rural China by the authors and their collaborators between 2012 and 2015. Of these, five surveys (Datasets 2, 3, 4, 6, and 7) were conducted in rural areas of Shaanxi and Gansu provinces in northwestern China. In terms of GDP per capita, Shaanxi and Gansu rank 14 and 31, respectively, out of China’s 31 provinces in 2017, Ref. [27] meaning that, at least in economic terms, they can be considered relatively representative of middle-income and low-income rural areas across China. The other survey (Dataset 5) was conducted among rural migrant students (children attending private schools for rural migrants) in Shanghai and Suzhou, two cities in eastern China. Therefore, the datasets collected by our research teams can be considered relatively comprehensive of children and adolescents in various rural communities of China.

The CFPS survey (Dataset 1) uses a three-stage “Probability-Proportional-to-Size” (PPS) sampling strategy with multistage stratification. First, 162 county-level units were randomly selected across 25 provinces. Second, 640 village-level units (villages in rural areas and residential communities in urban areas) were selected. Finally, 6317 households were selected from the village-level units. All members of each household who were at home during the time of the survey were interviewed, which included a total of 3057 children and adolescents within our target age range (9–16 years). The response rate was 84.1% at the individual level [28]. After excluding 378 children and adolescents who did not complete the mental health portion of the survey, the total number of children and adolescents included in this dataset is 2679.

The five surveys conducted by the research team in rural areas of China (Datasets 2, 3, 4, 6, and 7) used nearly identical four-step sampling strategies. First, members of the research team obtained a list of all counties in each of the sample provinces and selected counties from those that met the criteria for each study (e.g., large counties and counties that are known as nationally designated poverty counties). Second, using official records from each local county’s bureau of education, the research team created a list of all rural elementary and/or junior high schools in the sample counties. Third, the research team identified all schools that met a set of fixed characteristics (e.g., all elementary schools with grades one to six, schools primarily enrolling rural students, or schools with more than 100 students) and then randomly selected 755 schools from these lists (449 elementary schools and 306 junior high schools). Finally, within each sample school, we randomly selected sample classes in the target grades for each study and included all students in each sample class within the study sample.

To select the sample of rural migrant children (Dataset 5), the research team began by conducting a canvas-like survey to choose a sample of schools in each of the two sample cities. Unlike rural schools, however, no official lists of private migrant schools are available in China’s cities. We therefore contacted all educational and research institutes and non-profit organizations in the two cities that might have contact information for private migrant schools, and then called each school to confirm that it was still operating. In this way, we were able to establish a representative dataset of 94 private migrant schools in the two sample cities. We randomly selected one fifth grade class in each school and included all students in each chosen class within the sample. Nearly all children agreed to participate in the survey; however, 3236 children and adolescents (6.4% of the pooled sample) were excluded due to missing mental health outcomes or other missing data (355, 1232, 75, 322, 359, and 914 from Datasets 2 to 7, respectively). The final number of children in Datasets 2–7 is 50,742, including 27,103 elementary school students and 23,639 junior high school students.

Data from the CFPS survey were collected in 2012 in three blocks. The first block of the CFPS survey measured depression using the Center for Epidemiologic Studies Depression Scale (CES-D). The CES-D scale contains 20 items about the emotions experienced by the respondent within the past week, such as “I felt like my life is a failure,” and “I was bothered by things that usually do not bother me.” Survey participants were asked to respond using a Likert-type scale with four possible answers corresponding to how many days in the past week they had experienced the given emotion: “rarely or none of the time (less than 1 day)”, “some or a little of the time (1–2 days)”, “occasionally or a moderate amount of time (3–4 days)”, and “most or all of the time (5–7 days).” Possible scores range from 0 to 60, with scores of 17 or above indicating risk of depression [21].

The second block of the CFPS survey collected information on the cognition of sample children and adolescents using a numerical reasoning measure adapted from the University of Michigan Health and Retirement Survey (HRS) [29]. This test is a two-stage adaptive test in which each stage includes three items. Each item consists of a series of numbers with one number missing, and respondents are asked to fill in the number that fits the numerical pattern. In the first stage, all respondents were given the same three items. In the second stage, each respondent received test items of varying difficulty based on his or her performance in the first stage. Raw scores were standardized into a normal distribution with a mean of 0 and a standard deviation of 1 across the sample for this dataset.

The last block collected socioeconomic and demographic information from all sample children and adolescents. This included child gender, age in years, maternal and paternal education levels, and household wealth. In the CFPS survey, household socioeconomic status is measured by household net income.

Although the data collected by the authors were aggregated from multiple surveys, all six surveys used similar data collection strategies. In each survey, questionnaires were distributed by enumerators who underwent comprehensive, multi-day training.

In each of the six author surveys, the research team collected data in three blocks. The first block collected data on mental health outcomes. In Datasets 2–5, the mental health outcomes measured anxiety, whereas Datasets 6 and 7 assessed math anxiety specifically. To assess risk of anxiety, we used the Mental Health Test (MHT), which is derived from the Children’s Manifest Anxiety Scale (CMAS), an internationally standardized test of mood, anxiety, and other symptoms in children that has been used in the United States and other countries [13]. The MHT questionnaire contains 100 indicators for anxiety, to which respondents indicate whether they agree or disagree. Of the 100 questions, 10 are validity questions, while responses to the remaining 90 questions make up the MHT score. Although the test is not for diagnostic purposes, a higher score corresponds to a greater risk for anxiety, and a total score of 56 or above indicates a high risk and potential need for treatment. MHT results also can be broken down into eight anxiety subtypes, including (1) learning anxiety, (2) body anxiety, (3) self-blaming anxiety, (4) sensitivity tendency, (5) phobia anxiety, (6) social anxiety, (7) impulsive tendency, and (8) loneliness anxiety. A score above 8 on any subtype is considered clinically high and indicates that the respondent should seek further assessment and possible treatment.

Datasets 6 and 7 collected data on math anxiety from school-based surveys in rural China conducted by the authors. Math anxiety was assessed using a set of questions from the Program for International Student Assessment (PISA). PISA is a worldwide study conducted by the Organization of Economic Cooperation and Development (OECD) intended to evaluate educational systems in countries around the world. The math anxiety assessment in PISA includes a list of five statements, to which children and adolescents were asked to respond using a 4-point Likert-type scale of “strongly agree = 4,” “agree,” “disagree,” or “strongly disagree = 1”. The five statements are as follows: (a) “I often worry that math class will be difficult for me”, (b) “I get very tense when I have to do math homework”, (c) “I get very nervous doing math problems”, (d) “I feel helpless when doing a math problem”, and (e) “I worry that I will get poor grades in math” [30]. Scaled math anxiety scores were constructed by standardizing the raw scores into a mean of 0 and a standard deviation (SD) of 1 across each dataset. Math anxiety scores therefore represent the relative levels of math anxiety among children and adolescents within the sample, with positive scores indicating higher levels of math anxiety than the average for the sample.

The second block collected data on the standard math test scores of sample children and adolescents. We use scores on a standardized math test as a measure of standard math test scores among the sample. The math test in each experiment consisted of 25–30 questions, all of which were carefully chosen with the assistance of educators in the local education bureau of each sample area to ensure that test items were grade appropriate and consistent with the national curriculum. Within each dataset, raw scores were standardized into a normal distribution with a mean of 0 and a standard deviation (SD) of 1 across the sample.

The third block collected data on the basic demographic characteristics of all sample children and adolescents, including gender, age, grade level, maternal and paternal educational levels, and household wealth. To measure household wealth, we created a standardized family asset index using polychoric principal components analysis based on whether the family owned certain valuable items, such as a computer or car. We did so because recent studies suggest that using household asset indicators to construct continuous measures for household wealth may be more reliable than using self-reported income [14].

### 2.2. Ethics

The six surveys conducted by the authors and their colleagues were approved by the Stanford University Institutional Review Board. Study permissions were also obtained from the Chinese government. In accordance with IRB requirements, all children involved in the surveys provided oral assent for the project, and the school principals—who serve as the children’s legal guardians while they are in school—also provided their written consent.

### 2.3. Statistical Analysis

Using descriptive *t*-tests, we compare the prevalence of depression and anxiety, as well as standardized math anxiety scores, by demographic variables of sample children. We report the prevalences using the mean and standard deviation for consistency with the existing literature. Demographic variables include gender, grade level, household wealth, and parental education levels. In addition, we use ordinary least squares regression analysis to examine the correlations between depression, anxiety, or math anxiety and standard math test scores. The specification of the ordinary least squares regression analysis is below:(1)$$Cognition_{i}=\alpha_{0}+\alpha_{1}MH_{i}+X_{i}+\tau_{s}+\in_{i}$$
where $Cognition_{i}$ refers to standardized cognitive scores for child *i*, and $MH_{i}\;$is a dummy variable for depression, anxiety, or math anxiety, respectively. When the outcome is depression or anxiety, the dummy variable takes a value of 1 if child i is shows symptoms and 0 if not. When the outcome is math anxiety, the dummy variable takes the value of 1 if child i has a positive math anxiety score (indicating higher math anxiety than the average for the sample) and 0 if not. $X_{i}$ is a vector of control variables accounting for demographic characteristics of children and adolescents, including gender, age, household wealth, and parental education levels. $\tau_{s}$ represents strata fixed effects. When the outcome is depression, fixed effects are clustered at the village level for rural households and community level for urban households; when the outcome is anxiety or math anxiety, fixed effects are clustered at the school level.

## 3. Results

### 3.1. Descriptive Statistics

The characteristics of children and adolescents in our study are displayed in Table 2. The table includes data from both the authors own datasets and from the CFPS dataset. Overall, girls account for 48% of the sample. The average age of sample children is between 11 and 13 years old. Across all datasets, less than 40% of mothers and just over half of fathers graduated from junior high school.

The prevalence of risk for anxiety and depression among children and adolescents in our study is described in Table 3. The prevalence of depression (Dataset 1) is 20%. The prevalence of generalized anxiety is 6% in the combined dataset, and 8%, 5%, 6%, and 5% in Datasets 2, 3, 4, and 5, respectively. The prevalence of math anxiety is absent due to the lack of cut-off scores to screen for risk.

### 3.2. Prevalence of Depression and Anxiety by Demographic Characteristics

Table 4 compares the prevalence of risk for depression and anxiety by demographic characteristics. The results indicate that girls show a significantly greater risk than boys for depression and some types of anxiety symptoms, though the differences are relatively small in magnitude (Columns 1–3). Specifically, girls are 3% more likely to be at risk of depression (*p* = 0.03) and 2% more likely to be at risk of generalized anxiety (*p* < 0.001). Girls show a significantly higher risk for five out of eight subtypes of anxiety, including learning anxiety, body anxiety, self-blaming anxiety, phobias, and social anxiety (*p* < 0.001) for learning anxiety, self-blaming anxiety, phobias, and social anxiety; *p* = 0.01 for body anxiety). Girls are also 6% more likely to be at risk for any type of anxiety (*p* < 0.001).

The results show no significant differences in the prevalence of depression; however, risk for anxiety is significantly more prevalent among junior high school-aged adolescents (Table 4, Columns 4–6), though, once again, the differences are small. Specifically, adolescents in junior high school are 2% more likely to be at risk for generalized anxiety compared to children in elementary school (*p* < 0.001). Additionally, of the eight anxiety subtypes, adolescents in junior high school show a significantly higher risk for learning anxiety, body anxiety, self-blaming anxiety, sensitivity tendency, phobias, social anxiety, impulsive tendency, and loneliness anxiety (*p* = 0.01 for phobias and *p* < 0.001 for the others). Junior high schoolers are also 7% more likely to be at risk for any type of anxiety (*p* < 0.001).

The results also show that depression is negatively correlated with household wealth (Table 4, Columns 7–9). Children and adolescents from families with household wealth below the mean are 6% more likely to be at risk for depression (*p* < 0.001) compared to children from families with wealth above the mean. We also found significant but relatively small differences (no more than 2%) in five of the eight anxiety subcategories. Specifically, children and adolescents from families with below average wealth show a higher prevalence of sensitivity tendency, social anxiety, and impulsive tendency (all *p* < 0.001), whereas children and adolescents from families with above average wealth show a higher prevalence of self-blaming anxiety and phobias (*p* = 0.04 and *p* < 0.001, respectively). However, we found no significant differences in the prevalence of generalized anxiety or any specific single type of anxiety.

Children and adolescents whose parents have lower levels of education are at significantly greater risk for depression, yet the differences in anxiety are either small or insignificant (Table 4, Columns 10–12 and 13–15). Children and adolescents whose mothers and fathers have not graduated from junior high school are 8% and 9% more likely to be at risk for depression, respectively (both *p* < 0.001). Children and adolescents whose mothers or fathers did not complete junior high school are also 1%more likely to be at risk for generalized anxiety (both *p* = 0.01). Additionally, children and adolescents whose mothers did not complete junior high school show a higher prevalence of risk for learning anxiety, body anxiety, phobias, and any type of anxiety (*p* < 0.001, *p* < 0.001, *p* = 0.03, and *p* < 0.001, respectively). Similarly, children and adolescents whose fathers did not complete junior high school show a higher prevalence of risk for learning anxiety, body anxiety, and any anxiety (all *p* < 0.001).

Table 5 compares standardized math anxiety scores by demographic characteristics. The results show that girls and junior high school students have higher levels of math anxiety compared to boys and elementary school students, respectively. In contrast, children and adolescents with lower household wealth, and children and adolescents with lower parental education have lower levels of math anxiety than their higher-SES peers.

### 3.3. Correlations of Depression and Anxiety to Academic Performance

The correlations between depression, anxiety, and math anxiety and standard math test scores among sample children and adolescents are displayed in Table 6. The results show that of the 12 measures of depression and anxiety in our study, 10 are negatively correlated with standard math test scores. Depression is correlated with a decrease in cognitive scores by 0.32 SD (*p* < 0.001), while generalized anxiety is correlated with a decrease of 0.07 SD (*p* < 0.001). Learning anxiety, body anxiety, phobias, social anxiety, impulsive tendency, loneliness, and any type of anxiety are correlated with reduced cognitive scores by 0.12 SD, 0.16 SD, 0.16 SD, 0.06 SD, −0.01 SD, 0.24 SD, and 0.14 SD, respectively (*p* = 0.02 for social anxiety, *p* = 0.03 for impulsive tendency, and *p* < 0.001 for the others). Moreover, for each one-SD increase in math anxiety, cognitive scores for children and adolescents declined by 0.30 SD (*p* < 0.001). Only self-blaming anxiety is positively correlated with standard math test scores, with a coefficient of 0.06 SD (*p* < 0.001). Sensitivity tendency is not significantly correlated with standard math test scores.

## 4. Discussion

This study examined symptoms of depression and anxiety among children and adolescents across China, focusing on rural areas. Drawing on a pooled dataset of 53,421 school-aged children, the study described the overall prevalence of depression and anxiety, as well as the severity of math anxiety, and examined variation in the prevalence of symptoms among student subgroups. The study also examined the relationship between depressive and anxious symptoms and standard math test scores, controlling for demographic characteristics. The results found that 20% of children showed risk for depression, and 6% showed risk for generalized anxiety. Subgroup analyses revealed that girls and junior high school-aged children exhibit a greater prevalence of depression and anxiety symptoms and greater severity of math anxiety. Additionally, the results show robust and consistent negative correlations between the symptoms of depression and anxiety, as well as severity of math anxiety, and standard math test scores.

The results of this study reveal a high prevalence of depression and anxiety among Chinese children and adolescents compared to developed countries; however, the findings are similar to rates found in other developing countries. Our analysis using data from the nationally representative CFPS dataset found that 20% of children across China exhibit symptoms of depression, consistent with past studies in China and other middle-income countries. This is significantly higher than that of developed countries such as the United States, where studies have found a prevalence of 14.2% of depressive symptoms among children [28]. It is also higher than the global prevalence of depression among children, which is estimated to be 26% [4]. In contrast, our results are more consistent with findings in developing countries [31]. For example, a study among Egyptian children found that 23.8% showed depressive symptoms, and studies conducted in Nigeria have found the prevalence of children at risk for depression to be between 21.2% and 34.2%.

In addition, our results show that 6% of rural children and adolescents are at risk for anxiety (Table 2, Panel B). This is similar to previous studies in rural China: a study by Liu et al., (2018) [32] among rural students in five provinces across China found that 7% of students were at risk for anxiety. The prevalence of anxiety, broadly defined, in our study also is close to the average anxiety rate among students around the world. According to a systematic review of 87 studies across 44 countries, the prevalence of adolescent anxiety ranges from 6.5% to 7.2% [4]. However, it should be noted that this prevalence is higher than the adult Chinese population, which was 5.0% nationwide [33]. In addition, because anxiety is more likely to be stigmatized, it is possible that the actual prevalence of generalized anxiety is higher than past literature has reported, [34] especially in developing regions, where the stigma of mental illness is more prevalent [35].

Our data suggest that that the risk of depression and anxiety is more prevalent for certain subgroups of children and adolescents, including girls and older adolescents. Girls are at slightly higher risk than boys to develop almost all of the mental health problems examined in this study, including both depression and generalized anxiety, as well as specific forms of anxiety (learning anxiety, body anxiety, self-blaming anxiety, phobias, social anxiety, or any type of anxiety). Girls also have relatively higher levels of math anxiety than boys. Although no studies to date have comprehensively examined gendered differences in the prevalence of anxious and depressive symptoms among children and adolescents in rural China, our findings are supported by a series of international studies indicating that girls are generally more vulnerable to mental health problems than boys [8]. Although there may be biological contributors [36], these differences in vulnerability have, in part, been attributed to discrimination against girls through practices such as son preference, in which families grant higher value and better treatment to boys [9].

In addition, our results show that compared to elementary school-aged children, adolescents in junior high school show greater prevalence of depression and several anxiety symptoms, including higher levels of math anxiety. Although the results are once again relatively small, the differences are statistically significant, in contrast to research from other countries, which have found no significant differences between elementary school children and those in junior high school [37]. However, our findings are supported by several studies conducted in mainland China [32], which suggest that the highly competitive education system in China may adversely affect mental health as children and adolescents grow older. Whereas elementary school students can matriculate directly into junior high school, students in junior high school face tremendous pressure from the senior high school entrance exam. According to China’s Ministry of Education, the 2018 admittance rate for public senior high schools was 58%, meaning that over 40% of junior high school students in China were unable to attend academic high school and instead had to attend vocational high school or drop out of school entirely [38]. This is also supported by evidence from past studies of China’s college entrance exam, which have found that senior high school students tend to show increased anxiety because the exam can significantly affect future life outcomes [39].

Finally, our findings suggest that SES has different relations to mental health depending on the specific mental health problem examined. The data show that risk for depression and anxiety is more prevalent among children and adolescents with lower household wealth, and among children and adolescents whose mothers and fathers have lower levels of education, which is consistent with several recent studies that have found low SES to be predictive of depression and anxiety [16,40]. Maternal education, in particular, is found to relate to resilience from psychopathology, and the literature suggests that low maternal education is a proxy for risk of socioemotional problems among children [41]. In contrast, children and adolescents with higher SES, in terms of both household wealth and parental education levels, have higher math anxiety compared to those from lower SES subgroups. This contradicts the existing literature, which has overall demonstrated a negative correlation between SES and mental health problems [42]; it also contradicts the negative relation of parental education levels to learning anxiety in our sample. This may be explained by the correlation between parental out-migration and higher SES in rural China [43].

The results of our study indicate consistent correlations between depressive and anxiety symptoms and poor academic performance. The results show that depression, generalized anxiety, learning anxiety, body anxiety, phobias, social anxiety, impulsive anxiety, loneliness, any type of anxiety, and math anxiety are all significantly correlated with lower levels of math test scores among sample children and adolescents. Although the magnitudes of correlation are small, the results are consistent with previous studies, which have similarly shown that depression and anxiety are strong determinants of poor academic performance in both developed and developing countries [44].

This study has a number of strengths. First, our aggregated sample of seven datasets (N = 53,421) is much larger than samples used in previous studies, giving it a high degree of statistical power and external validity. Second, observations for the same panel of data (depression, anxiety, or math anxiety) were collected by a single research team using a common sampling strategy as well as standardized data collection instruments and enumeration processes, allowing us to compare across datasets and subgroups.

Our study also has two limitations. First, the study instruments used to assess depression and anxiety are self-reported screening tools, which can identify symptoms but cannot diagnose a psychiatric disorder. Second, this study only included cross-sectional data; as such, we are unable to explore causal relations among risk factors that predict long-term outcomes. Future research should collect longitudinal cohort data to understand causal mechanisms underlying the relations identified in this study.

## 5. Conclusions

Children and adolescents in LMICs account for almost 90% of the global population under 18 years of age. However, only 10% of the literature on pediatric mental health is representative of these youth, and even fewer studies come from rural populations. As China constitutes 15% of the global population of children under age 18, this study offers valuable information to the field of global mental health, especially given its findings of high prevalence for multiple mental health problems and robust correlations between mental health and academic performance. Our findings suggest that certain subgroups, including girls, students in higher grade levels, and students with lower SES, may be more susceptible to depression and anxiety, which may inform future targeted interventions to improve child and adolescent mental health and wellbeing. Taken together, these findings call for urgent political and public health action to implement contextually informed mental health prevention and treatment programs for youth in rural China and in underdeveloped communities around the world.

## Figures and Tables

**Table 1 ijerph-19-05026-t001:** Distribution and location of datasets.

	Dataset	Date	Location	Grade	Clusters	Obs.	Authorship of the Major Publications
Panel A: Depression	Dataset 1	2012	Nationwide	10–15 yrs. old	640 villages/communities	2679	Mi Zhou, Guangsheng Zhang, Scott Rozelle, Kaleigh Kenny, and Hao Xue (2018) [21]
Panel B: Anxiety (MHT)	Dataset 2	2012	Shaanxi	7 and 8	75 schools	8915	Huan Wang, Yang Chu, Fei He, Yaojiang Shi, Qinghe Qu, Scott Rozelle, and James Chu (2014) [22]
	Dataset 3	2013	Shaanxi/Gansu	4 and 5	252 schools	19,363	Hongyu Guan, Huan Wang, Kang Du, Jin Zhao, Matthew Boswell, Yaojiang Shi, and Yiwei Qian (2018) [23]
	Dataset 4	2013	Shaanxi	7 and 8	31 schools	2020	Jingchun Nie, Qian Zhou, Hong Ouyang, Jiayuan Gao, Lei Tang (2019) [24]
	Dataset 5	2013	Shanghai/Suzhou	5	94 schools	4260	Qiran Zhao, Xiaobing Wang, Scott Rozelle (2019) [25]
Panel C: Math Anxiety	Dataset 6	2014	Shaanxi/Gansu	4	103 schools	3480	Meichen Lu (2018) [26]
	Dataset 7	2015	Shaanxi/Gansu	7	200 schools	12,704	

**Table 2 ijerph-19-05026-t002:** Child/adolescent demographic characteristics by mental health outcome.

	Panel A Depression	Panel B Anxiety					Panel C Math Anxiety		
	Dataset 1 (*n* = 2679)	Combined (*n* = 34,558)	Dataset 2 (*n* = 8915)	Dataset 3 (*n* = 19,363)	Dataset 4 (*n* = 2020)	Dataset 5 (*n* = 4260)	Combined (*n* = 16,184)	Dataset 6 (*n* = 3474)	Dataset 7 (*n* = 12,560)
Gender (1 = female)	0.48	0.47	0.46	0.48	0.47	0.45	0.49	0.48	0.49
	(0.50)	(0.50)	(0.50)	(0.50)	(0.50)	(0.50)	(0.50)	(0.50)	(0.50)
Age (years)	12.60	11.49	13.48	10.50	13.56	10.98	12.57	10.82	13.05
	(1.68)	(1.69)	(1.14)	(0.97)	(1.14)	(0.81)	(1.33)	(0.94)	(0.97)
Household wealth above average, mean (SD)	0.39	0.34	0.48	0.27	--	0.34	0.50	0.49	0.50
	(0.49)	(0.47)	(0.50)	(0.44)	--	(0.47)	(0.50)	(0.50)	(0.50)
Mother education is junior high or above, mean (SD)	0.41	0.36	0.26	0.35	0.26	0.48	0.35	0.36	0.34
	(0.49)	(0.48)	(0.44)	(0.48)	(0.44)	(0.50)	(0.48)	(0.48)	(0.47)
Father education is junior high or above, mean (SD)	0.56	0.48	0.45	0.54	0.46	0.67	0.56	0.56	0.57
	(0.50)	(0.50)	(0.50)	(0.50)	(0.50)	(0.47)	(0.50)	(0.50)	(0.50)

Data Source: Author and Collaborators’ Surveys, and China Family Panel Study (CFPS) 2012 Survey on *Prevalence of Anxiety and Depression.*

**Table 3 ijerph-19-05026-t003:** Prevalence of depression and anxiety (presented in mean and SD).

	Mental Health Outcome	Dataset 1	Combined	Dataset 2	Dataset 3	Dataset 4	Dataset 5
Panel A: Depression	Depression	0.20					
		(0.40)					
Panel B: Anxiety	Generalized anxiety		0.06	0.08	0.05	0.06	0.05
			(0.24)	(0.27)	(0.22)	(0.24)	(0.23)
	Subcategories						
	Learning anxiety		0.59	0.64	0.56	0.65	0.60
			(0.49)	(0.48)	(0.50)	(0.48)	(0.49)
	Body anxiety		0.19	0.22	0.18	0.24	0.18
			(0.40)	(0.41)	(0.39)	(0.43)	(0.38)
	Self-blaming anxiety		0.18	0.19	0.17	0.24	0.21
			(0.39)	(0.39)	(0.37)	(0.43)	(0.40)
	Sensitivity tendency		0.13	0.17	0.10	0.20	0.13
			(0.33)	(0.37)	(0.30)	(0.40)	(0.33)
	Phobias		0.11	0.10	0.11	0.16	0.10
			(0.31)	(0.31)	(0.31)	(0.37)	(0.29)
	Social anxiety		0.07	0.08	0.06	0.13	0.06
			(0.25)	(0.27)	(0.23)	(0.33)	(0.23)
	Impulsive tendency		0.03	0.04	0.02	0.07	0.03
			(0.16)	(0.19)	(0.13)	(0.25)	(0.17)
	Loneliness		0.02	0.02	0.02	0.08	0.02
			(0.15)	(0.15)	(0.13)	(0.26)	(0.14)
	Any anxiety		0.68	0.73	0.65	0.72	0.69
			(0.47)	(0.44)	(0.48)	(0.45)	(0.46)

Data source: Authors and collaborators’ surveys, and China Family Panel Data (CFPS) 2012.

**Table 4 ijerph-19-05026-t004:** Prevalence of depression and anxiety by demographic characteristics (presented in mean and SD).

	(1) Gender			(2) Grade			(3) Household Wealth			(4) Maternal Education			(5) Paternal Education		
	Girls	Boys	Diff	Elementary	Jhs ^1^	Diff	≥Avg	<Avg	Diff	≥Jhs	<Jhs	Diff	≥Jhs	<Jhs	Diff
Depession	0.22	0.19	0.03 *	0.19	0.21	−0.02	0.17	0.23	−0.06 **	0.15	0.23	−0.08 **	0.16	0.25	−0.09 **
	(0.41)	(0.39)		(0.40)	(0.41)		(0.37)	(0.42)		(0.36)	(0.42)		(0.37)	(0.43)	
Anxiety															
Generalized anxiety	0.06	0.04	0.02 **	0.05	0.07	−0.02 **	0.06	0.06	0.00	0.05	0.06	−0.01 **	0.05	0.06	−0.01 **
	(0.25)	(0.22)		(0.22)	(0.26)		(0.24)	(0.23)		(0.23)	(0.24)		(0.22)	(0.24)	
Subcategories															
Learning anxiety	0.62	0.57	0.05 **	0.57	0.64	−0.07 **	0.59	0.59	0.00	0.57	0.61	−0.04 **	0.58	0.60	−0.02 **
	(0.48)	(0.50)		(0.50)	(0.48)		(0.49)	(0.49)		(0.50)	(0.49)		(0.49)	(0.49)	
Body anxiety	0.20	0.19	0.01 **	0.18	0.22	−0.04 **	0.20	0.19	0.01	0.18	0.20	−0.02 **	0.18	0.20	−0.02 **
	(0.40)	(0.39)		(0.39)	(0.41)		(0.40)	(0.39)		(0.38)	(0.40)		(0.39)	(0.40)	
Self-blaming anxiety	0.22	0.15	0.07 **	0.17	0.20	−0.03 **	0.17	0.18	−0.01 *	0.17	0.18	−0.01 *	0.18	0.18	0.00
	(0.41)	(0.36)		(0.38)	(0.40)		(0.38)	(0.39)		(0.38)	(0.39)		(0.38)	(0.38)	
Sensitivity tendency	0.13	0.13	0.00	0.11	0.17	−0.06 **	0.14	0.12	0.02 **	0.12	0.13	−0.01	0.12	0.13	−0.01
	(0.34)	(0.33)		(0.31)	(0.38)		(0.34)	(0.32)		(0.33)	(0.33)		(0.33)	(0.33)	
Phobias	0.16	0.06	0.10 **	0.10	0.11	−0.01 **	0.10	0.11	−0.01 **	0.10	0.11	−0.01 *	0.10	0.11	−0.01
	(0.36)	(0.24)		(0.31)	(0.32)		(0.30)	(0.31)		(0.30)	(0.31)		(0.30)	(0.31)	
Social anxiety	0.07	0.06	0.01 **	0.06	0.09	−0.03 **	0.07	0.06	0.01 **	0.06	0.07	−0.01	0.06	0.07	−0.01
	(0.26)	(0.24)		(0.23)	(0.29)		(0.25)	(0.24)		(0.24)	(0.25)		(0.24)	(0.25)	
Impulsive tendency	0.03	0.03	0.00	0.02	0.04	−0.02 **	0.03	0.02	0.01 **	0.02	0.02	0.00	0.02	0.02	0.00
	(0.17)	(0.16)		(0.14)	(0.20)		(0.18)	(0.15)		(0.15)	(0.15)		(0.15)	(0.16)	
Loneliness	0.02	0.02	0.00	0.02	0.03	−0.01 **	0.02	0.02	0.00	0.02	0.02	0.00	0.02	0.02	0.00
	(0.14)	(0.15)		(0.13)	(0.18)		(0.13)	(0.14)		(0.13)	(0.14)		(0.13)	(0.14)	
Any anxiety	0.71	0.65	0.06 **	0.66	0.73	−0.07 **	0.67	0.68	−0.01	0.65	0.69	−0.04 **	0.66	0.69	−0.03 **
	(0.45)	(0.48)		(0.48)	(0.45)		(0.47)	(0.47)		(0.48)	(0.46)		(0.47)	(0.46)	

Data source: Author and their collaborators’ surveys, and China Family Panel Data (CFPS) 2012. Note: The cluster-robust standard errors of each regression are presented in the parentheses; * *p* < 0.05; ** *p* < 0.01.

**Table 5 ijerph-19-05026-t005:** Comparison of math anxiety standardized scores by demographic characteristics.

	(1) Gender			(2) Grade			(3) Household Wealth			(4) Mother’s Education			(5) Father’s Education		
	Girls	Boys	Diff	Elementary	Jhs ^1^	Diff	≥Avg	<Avg	Diff	≥Jhs	<Jhs	Diff	≥Jhs	<Jhs	Diff
Math anxiety	0.04	−0.04	0.08 **	−0.06	0.02	−0.08 **	0.02	−0.02	0.04 **	0.02	−0.05	0.07 **	0.04	−0.03	0.07 **
	(0.65)	(0.65)		(0.63)	(0.65)		(0.65)	(0.65)		(0.65)	(0.65)		(0.65)	(0.64)	

Data source: Author surveys. Note: ** *p* < 0.01. ^1^ Jhs refers to junior high school.

**Table 6 ijerph-19-05026-t006:** Correlations between math test scores (dependent variable) and student mental health (independent variables).

	Depression	Generalized Anxiety	Learning Anxiety	Body Anxiety	Self-Blaming Anxiety	Sensitivity Tendency	Phobias	Social Anxiety	Impulsive Tendency	Loneliness	Any Anxiety	Math Anxiety
Scores	−0.317 **	−0.067 **	−0.124 **	−0.155 **	0.059 **	0.016	−0.155 **	−0.057 *	−0.009 **	−0.244 **	−0.135 **	−0.300 **
	(0.077)	(0.023)	(0.014)	(0.016)	(0.016)	(0.019)	(0.016)	(0.024)	(0.016)	(0.039)	(0.014)	(0.013)
Demographic characteristics	Yes	Yes	Yes	Yes	Yes	Yes	Yes	Yes	Yes	Yes	Yes	Yes
School fixed effects	Yes	Yes	Yes	Yes	Yes	Yes	Yes	Yes	Yes	Yes	Yes	Yes
N	2679	34,558	34,558	34,558	34,558	34,558	34,558	34,558	34,558	34,558	34,558	16,184
adj. R-square	0.089	0.014	0.016	0.019	0.016	0.016	0.019	0.016	0.019	0.017	0.020	0.104

Data source: Author and their collaborators’ surveys, and China Family Panel Data (CFPS) 2012. Note: The cluster-robust standard errors of each regression are presented in the parentheses; * *p* < 0.05; ** *p* < 0.01.

## Data Availability

Data are available upon request.

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
