# Peer review of "Depressive and Anxiety Symptoms among Children and Adolescents in Rural China: A Large-Scale Epidemiological Study"

_ijerph, 2022, doi:10.3390/ijerph19095026_

Round 1
Reviewer 1 Report
General comments and recommendations
The title of the manuscript and the text refer to the mental health of children and adolescents. However, the variables evaluated are limited to internalizing problems or disorders (depression and anxiety), so the title is misleading and inappropriate. It should be changed.
The manuscript can only be properly understood when relevant information is found in the annexes. The authors should consider including part of this information in the article itself, especially that related to the Method: general procedure, sampling method in the datasets and evaluation instruments (questionnaires) included in the surveys. In any case, the information in the annexes is necessary.
The study does not work with a single sample but with a set of data from six surveys carried out by the authors' research team. This procedure is unusual, but it is adequately explained (especially in the annexes) and the data comes from a very large number of children. The honesty of the authors in providing this information is valued positively.
Specific recommended changes:
Lines 64-66: “For example, some studies have found that that girls are more likely to have anxiety and depression than boys,[11] whereas other studies have found no gender differences in either anxiety or depression.”
What are those other studies? It is necessary to indicate the reference or references with their numbers at the end of the sentence.
Line 95/ Table 1: The major publications included in this table (last column) should be listed in the References section. It would be interesting to have the name of the authors of these publications in that column, which would facilitate the verification that they are publications of the research team that submits the manuscript.
Results / Table 4: The differences in prevalence risk, even when they are significant, are not very large in general. There are significant differences of 0.01 (1%) (eg body anxiety /gender), 0.02, 0.03, ... exceptionally 0.10 (10%) in phobias (gender). It is necessary to take this into account when discussing the differences in prevalence risk, since in many cases they are very small.
Results / Table 5: Similarly, the correlations are very low even though they are significant. To call these correlations "robust" (Discussion) is an exaggeration.
(The lines are missing in the Discussion section of the manuscript, pages 13 to 15).
Discussion, page 13, paragraph 3. "This is similar to previous studies in rural China: a study by Liu et al. (2018) among rural students...". The reference number is missing: is it 22?
Discussion, page 13, last paragraph. “Our data suggest that that the risk of mental health problems is especially serious for certain subgroups of children and adolescents, including girls and older adolescents.” "Especially serious" with the differences so small that I have already commented, seems exaggerated.
Discussion, page 14, paragraph 2. “In addition, our results show that compared to elementary school-age children, adolescents in junior high school are at greater risk for mental health problems, including higher levels of math anxiety."It should also be tempered.
Author Response
Reviewer #1
General Comment 1: The title of the manuscript and the text refer to the mental health of children and adolescents. However, the variables evaluated are limited to internalizing problems or disorders (depression and anxiety), so the title is misleading and inappropriate. It should be changed.
Response to general comment 1: In the revised manuscript, we have changed the title to “Depressive and anxiety symptoms among children and adolescents in rural China: A large-scale epidemiological study”. We have also revised the text throughout the manuscript to refer more specifically to symptoms of depression and anxiety, rather than mental health generally. All revisions have been made with tracked changes for the reviewers’ convenience.
General Comment 2: The manuscript can only be properly understood when relevant information is found in the annexes. The authors should consider including part of this information in the article itself, especially that related to the Method: general procedure, sampling method in the datasets and evaluation instruments (questionnaires) included in the surveys. In any case, the information in the annexes is necessary.
Response to general comment 2: Thank you for this comment. In our revised manuscript, we have incorporated the annexes into the main body of the manuscript.
General Comment 3: The study does not work with a single sample but with a set of data from six surveys carried out by the authors' research team. This procedure is unusual, but it is adequately explained (especially in the annexes) and the data comes from a very large number of children. The honesty of the authors in providing this information is valued positively.
Response to general comment 3: Thank you for your thoughtful consideration of the methods and sample used in this study.
Followings are a number of specific recommended changes:
Comment 1: Lines 64-66: “For example, some studies have found that that girls are more likely to have anxiety and depression than boys,[11] whereas other studies have found no gender differences in either anxiety or depression.” What are those other studies? It is necessary to indicate the reference or references with their numbers at the end of the sentence.
Response to comment 1: Thank you for noticing. We have added the corresponding citations. The missing citations were no. 12-13 in our original manuscript and become no.19-20 in our revised manuscript.
Line 64-66:
“For example, some studies have found that girls are more likely to have anxiety and depression than boys,[18] whereas other studies have found no gender differences in either anxiety or depression. [19-20]”
The added citations refer to the following two publications:
- Chen, Jie, et al. "Genetic and environmental contributions to anxiety among Chinese children and adolescents–a multi‐informant twin study." Journal of Child Psychology and Psychiatry 56.5 (2015): 586-594.
- Tepper, Ping, et al. "Depressive symptoms in Chinese children and adolescents: parent, teacher, and self reports." Journal of affective disorders 111.2-3 (2008): 291-298.
Comment 2: Line 95/ Table 1: The major publications included in this table (last column) should be listed in the References section. It would be interesting to have the name of the authors of these publications in that column, which would facilitate the verification that they are publications of the research team that submits the manuscript.
Response to comment 2: In response to this comment, we have moved the publications to the references list, and we have revised Table 2 to include the author names for all publications.
Comment 3: Results / Table 4: The differences in prevalence risk, even when they are significant, are not very large in general. There are significant differences of 0.01 (1%) (eg body anxiety /gender), 0.02, 0.03, ... exceptionally 0.10 (10%) in phobias (gender). It is necessary to take this into account when discussing the differences in prevalence risk, since in many cases they are very small.
Response to comment 3: Thank you for this comment. In our revised manuscript, we have clarified that many of the differences in prevalence are relatively small despite their statistical significance.
Line 167-175:
“Table 4 compares the prevalence of risk for depression and anxiety by demo-graphic characteristics. The results indicate that girls show significantly greater risk than boys for depression and some types of anxiety symptoms, though the differences are relatively small in magnitude (Columns 1-3). Specifically, girls are 3 per-centage points more likely to be at risk of depression (P=0·03) and 2 percentage points more likely to be at risk of generalized anxiety (P<0·001). Girls show a significantly higher risk for five out of eight subtypes of anxiety, including learning anxiety, body anxiety, self-blaming anxiety, phobias, and social anxiety (P<0·001) for learning anxiety, self-blaming anxiety, phobias, and social anxiety; P=0·01 for body anxiety). Girls are also 6 percentage points more likely to be at risk for any type of anxiety (P<0·001).
The results show no significant differences in the prevalence of depression; however, risk for anxiety is significantly more prevalent among junior high school-age adolescents (Table 4, Columns 4-6), though, once again, the differences are small. Specifically, adolescents in junior high school are 2 per-centage points more likely to be at risk for generalized anxiety compared to children in elementary school (P<0·001). Additionally, of the eight anxiety subtypes, adolescents in junior high school show a significantly higher risk for learning anxiety, body anxiety, self-blaming anxiety, sensitivity tendency, phobias, social anxiety, impulsive tendency, and loneliness anxiety (P=0·01 for phobias and P<0·001 for the others). Junior high schoolers are also 7 percentage points more likely to be at risk for any type of anxiety (P<0·001).
The results also show that depression is negatively correlated with household wealth (Table 4, Columns 7-9). Children and adolescents from families with household wealth below the mean are 6 percentage points more likely to be at risk for depression (P<0·001) compared to children from families with wealth above the mean. We also find significant but relatively small differences (no more than 2 percentage points) in five of the eight anxiety subcategories. Specifically, children and adolescents from families with below average wealth show a higher prevalence of sensitivity tendency, social anxiety and impulsive tendency (all P<0·001), whereas children and adolescents from families with above average wealth show a higher prevalence of self-blaming anxiety and phobias (P=0·04, P<0·001, respectively). However, we find no significant differences in the prevalence of generalized anxiety or any specific single type of anxiety.
Children and adolescents whose parents have lower levels of education are at significantly greater risk for depression, yet the differences in anxiety are either small or insignificant (Table 4, Columns 10-12 and 13-15). Children and adolescents whose mothers and fathers have not graduated from junior high school are 8 and 9 percentage points more likely to be at risk for depression, respectively (both P<0·001). Children and adolescents whose mothers or fathers have not completed junior high school are also 1 percentage point more likely to be at risk for generalized anxiety (both P=0·01). Additionally, children and adolescents whose mothers have not completed junior high school show a higher prevalence of risk for learning anxiety, body anxiety, phobias, and any type of anxiety (P<0·001, P<0·001, P=0·03, P<0·001, respectively). Similarly, children and adolescents whose fathers have not completed junior high school show a higher prevalence of risk for learning anxiety, body anxiety and any anxiety (all P<0·001).”
Comment 4: Results / Table 5: Similarly, the correlations are very low even though they are significant. To call these correlations "robust" (Discussion) is an exaggeration.
Response to comment 4: We acknowledge some coefficients of the regressions are small. We have removed the term “robust” from our discussion of the correlations in our revised manuscript.
Paragraph 7, Discussion section:
“The results of our study indicate consistent correlations between depressive and anxiety symptoms and poor academic performance. The results show that depression, generalized anxiety, learning anxiety, body anxiety, phobias, social anxiety, impulsive anxiety, loneliness, any type of anxiety, and math anxiety, are all significantly correlated with lower levels of math test scores among sample children and adolescents. Although the magnitudes of correlation are small, the results are consistent with previous studies, which have similarly shown that depression and anxiety are strong determinants of poor academic performance in both developed and developing countries.[30]”
(from Reviewer #1: The lines are missing in the Discussion section of the manuscript, pages 13 to 15).
Comment 5: Discussion, page 13, paragraph 3. "This is similar to previous studies in rural China: a study by Liu et al. (2018) among rural students...". The reference number is missing: is it 22?
Response to comment 5: Thank you for noticing. The missing citation is indeed citation 22. In our revised manuscript, we have fixed this typo.
Comment 6: Discussion, page 13, last paragraph. “Our data suggest that that the risk of mental health problems is especially serious for certain subgroups of children and adolescents, including girls and older adolescents.” "Especially serious" with the differences so small that I have already commented, seems exaggerated.
Response to comment 6: We agree with the reviewer that the conclusion may be exaggerated. We have revised our discussion section in response to this comment.
Page 13, last paragraph:
“Our data suggest that that the risk of mental health problems is more prevalent for certain subgroups of children and adolescents, including girls and older adolescents. Girls are at slightly higher risk than boys to develop almost all of the mental health problems examined in this study, including both depression and generalized anxiety, as well as specific forms of anxiety (learning anxiety, body anxiety, self-blaming anxiety, phobias, social anxiety, or any type of anxiety). Girls also have relatively higher levels of math anxiety than boys. Although no studies to date have comprehensively examined gendered differences in the prevalence of anxious and depressive symptoms among children and adolescents in rural China, our findings are supported by a series of international studies indicating that girls are generally more vulnerable to mental health problems than boys.[6] Although there may be biological contributors,[20] these differences in vulnerability have, in part, been attributed to discrimination against girls through practices such as son preference, in which families grant higher value and better treatment to boys.[7]”
Comment 7: Discussion, page 14, paragraph 2. “In addition, our results show that compared to elementary school-age children, adolescents in junior high school are at greater risk for mental health problems, including higher levels of math anxiety." It should also be tempered.
Response to comment 7: In our revised manuscript, we have tempered the wording of our conclusions.
Page 14, second paragraph:
“In addition, our results show that compared to elementary school-age children, adolescents in junior high school show greater prevalence of depression and several anxiety symptoms, including higher levels of math anxiety. Although the results are once again relatively small, the differences are statistically significant, in contrast to research from other countries, which have found no significant differences between elementary school children and those in junior high school.[21] However, our findings are supported by several studies conducted in mainland China,[22] which suggest that the highly competitive education system in China may adversely affect mental health as children and adolescents grow older. Whereas elementary school students can matriculate directly into junior high school, students in junior high school face tremendous pressure from the Senior High School Entrance Exam. According to China’s Ministry of Education, the 2018 admittance rate for public senior high schools was 58%, meaning that over 40% of junior high school students in China were unable to attend academic high school and instead had to attend vocational high school or drop out of school entirely.[23] This is also supported by evidence from past studies of China’s college entrance exam, which have found that senior high school students tend to show in-creased anxiety because the exam can significantly affect future life out-comes.[24]”
The evaluation form of reviewer #1:
Response to the evaluation form: We have incorporated all the appendices into the main body of the text to improve the transparency of the study design and methods. We would be happy to further revise our manuscript if the reviewer has specific comments on the design and the methods of our study.

Reviewer 2 Report
This paper analyzes combined data from seven cohorts of children in rural China regarding mental health and math anxiety collected 2012-2015.
The following are our comments:
- The appendices should be placed in the main text.
- The authors do not discuss the participation rate. How many / what % of students declined to participate? Similarly, how were missing data handled?
- Cluster-robust standard errors are used in many of the tables in the papers cited presenting the original cohort data, but the authors here use standard deviations. If standard deviations are felt to be the appropriate measure, a justification in the methods section would be helpful.
- Are there any temporal trends, since the cohorts were apparently assembled over a period of several years?
- Table 2:
- We don’t think the note is necessary since it is mentioned in the text
- The formatting could be better – maybe instead of saying 1=*category*, include the reference value as well. Also formatting as Gender, mean (SD) (for example)
- Put observation # at top, and use commas
- Table 3: remove Note – it is already stated in text
- Table 4: include math anxiety, not sure why it is in appendix
- Table 5: needs better labeling. Are the numbers in parentheses standard deviations? If so, this should be stated in the title or footnote.
- Line 220-224, “impulsive tendency” has the wrong value. We think it should be .01. "1 Learning anxiety, 2 body anxiety, 3 phobias, 4 social anxiety, 5 impulsive tendency, 6 loneliness, and 7 any one type of anxiety are correlated with reduced cognitive scores by 1 0·12 SD, 2 0·16 SD, 3 0·16 SD, 4 0·06 SD, 5 0·03 SD, 6 0·24 SD, and 7 0·14 SD, respectively (P=0·02 for social anxiety, P=0·03 for impulsive tendency, and P<0.001 for the others). “ If the P value is correct as .03, we think it also should have one asterisk mark, not two.
- Discussion: The bottom half of the first paragraph could be moved to the results.
Author Response
Reviewer #2
This paper analyzes combined data from seven cohorts of children in rural China regarding mental health and math anxiety collected 2012-2015.
The following are our comments:
Comment 1: The appendices should be placed in the main text.
Response to comment 1: Thank you for your comment. In our revised manuscript, we have incorporated all appendices into the main manuscript.
Comment 2: The authors do not discuss the participation rate. How many / what % of students declined to participate? Similarly, how were missing data handled?
Response to comment 2: We acknowledge that it is important to include the participation rate among students who were sampled for this study, as well as describe our methods for addressing missing data. For the CFPS dataset (dataset 1), the response rate was 84.14% at the individual level at the baseline survey. We excluded 378 children and adolescents (12.4% of the total sample) who did not complete the mental health portion of the survey. For the other datasets, because the surveys were completed under the organization of teachers at the local schools, the completion rate were nearly 100% across dataset 2 to 7. We excluded 3,236 children and adolescents who did not have the mental health outcomes or missed other values (355, 1,232, 75, 322, 359, 914 from Dataset 2 to 7, respectively), which account for 6.4% of the total sample. In our revised manuscript, we have clarified the related information.
Page 11, the fourth paragraph:
“The CFPS survey (Dataset 1) uses a three-stage “Probability-Proportional-to-Size” (PPS) sampling strategy with multistage stratification. First, 162 county-level units were randomly selected across 25 provinces. Second, 640 village-level units (villages in rural areas and residential communities in urban areas) were selected. Finally, 6,317 households were selected from the village-level units. All members of each household who were at home during the time of the survey were interviewed, which includes a total of 3,057 children and adolescents within our target age range (9-16 years). The individual response rate was 84.1%.[21] After excluding 378 children and adolescents who did not complete the mental health portion of the survey, the total number of children and adolescents included in this dataset is 2,679.
The five surveys conducted by the research team in rural areas of China (Datasets 2, 3, 4, 6 & 7) used nearly identical four-step sampling strategies. First, members of the research team obtained a list of all counties in each of the sample provinces and selected counties from those that met the criteria for each study (e.g., large counties; counties that are known as nationally-designated poverty counties). Second, using official records from each local county’s bureau of education, the research team created a list of all rural elementary and/or junior high schools in the sample counties. Third, the research team identified all schools that met a set of fixed characteristics (e.g., all elementary schools with grades one to six; schools primarily enrolling rural students; or schools with more than 100 students) and then randomly selected 755 schools from these lists (449 elementary schools and 306 junior high schools). Finally, within each sample school, we randomly selected sample classes in the target grades for each study and included all students in each sample class within the study sample.
To select the sample of rural migrant children (Dataset 5), the research team began by conducting a canvas-like survey to choose a sample of schools in each of the two sample cities. Unlike rural schools, however, no official lists of private migrant schools are available in China’s cities. We therefore contacted all educational and research institutes and non-profit organizations in the two cities that might have contact information for private migrant schools, and then called each school to confirm that it was still operating. In this way, we were able to establish a representative dataset of 94 private migrant schools in the two sample cities. We randomly selected one fifth grade class in each school and included all students in each chosen class within the sample. The total number of children in Datasets 2-7 is 50,742, including 27,103 elementary school students and 23,639 junior high school students. For datasets 2-7, almost no students refused to participate in the surveys, and no students were excluded due to missing mental health outcomes.”
Comment 3: Cluster-robust standard errors are used in many of the tables in the papers cited presenting the original cohort data, but the authors here use standard deviations. If standard deviations are felt to be the appropriate measure, a justification in the methods section would be helpful.
Response to comment 3: Thank you for your comment. We acknowledge the cluster-robust standard errors were reported for all regression analyses in the papers included in Table 1. In the present study, we have similarly included cluster-robust standard errors for our regression analysis in Table 5. However, in Tables 2 through 4, where we report prevalence and simple comparisons only (rather than regression analyses), we chose to include the means and standard deviations as they are the most common parameters for prevalence estimates in the literature. We believe that using mean and standard deviation for reporting prevalence allows our results to be more easily comparable to prevalences reported in other publications. We have clarified this point in our revised manuscript.
Line 139-147:
“Using descriptive t-tests, we compare the prevalence of depression and anxiety, as well as standardized math anxiety scores, by demographic variables of sample children. We report prevalence using mean and standard deviation for consistency with the existing literature. Demographic variables include gender, grade level, household wealth, and parental education levels. In addition, we use ordinary least squares regression analysis to examine the correlations between mental health problems and standard math test scores while controlling for other potential confounders including gender, age, household wealth, and parental education levels.”
Comment 4: Are there any temporal trends, since the cohorts were apparently assembled over a period of several years?
Response to comment 4: Thank you for this interesting question. We chose not to compare the temporal trends as there is limited temporal variation across the samples. For Panel B, in which the same outcomes were measured in 4 datasets, three were measured in 2012 and one in 2013. This means there is only a temporal difference of one year. Similarly, for Panel C, in which the same outcomes were measured in 2 datasets, one was measured in 2014 and the other in 2015. We believe the one-year difference will make the temporal comparison vulnerable to other confounding factors, so we have chosen not to include temporal comparisons.
Comment 5: Table 2:
We don’t think the note is necessary since it is mentioned in the text
The formatting could be better – maybe instead of saying 1=*category*, include the reference value as well. Also formatting as Gender, mean (SD) (for example)
Put observation # at top, and use commas
Response to comment 5: In our revised manuscript, we have updated Table 2 per this comment.
Comment 6: Table 3: remove Note – it is already stated in text
Response to comment 6: The note for Table 3 has been deleted in the revised manuscript.
Comment 7: Table 4: include math anxiety, not sure why it is in appendix
Response to comment 7: In our revised manuscript, we have incorporated all appendices into the main manuscript.
Comment 8: Table 5: needs better labeling. Are the numbers in parentheses standard deviations? If so, this should be stated in the title or footnote.
Response to comment 8: The numbers in the parentheses are cluster-robust standard errors. We have added a note in the bottom of Table 5 to clarify this in our revised manuscript. We have also clarified this point in our revised methods section.
Comment 9: Line 220-224, “impulsive tendency” has the wrong value. We think it should be .01. "1 Learning anxiety, 2 body anxiety, 3 phobias, 4 social anxiety, 5 impulsive tendency, 6 loneliness, and 7 any one type of anxiety are correlated with reduced cognitive scores by 1 0·12 SD, 2 0·16 SD, 3 0·16 SD, 4 0·06 SD, 5 0·03 SD, 6 0·24 SD, and 7 0·14 SD, respectively (P=0·02 for social anxiety, P=0·03 for impulsive tendency, and P<0.001 for the others). “ If the P value is correct as .03, we think it also should have one asterisk mark, not two.
Response to comment 9: Thank you for noticing this typo. The coefficient for impulsive tendency should be -0.01. We have corrected this typo in our revised manuscript.
Comment 10: Discussion: The bottom half of the first paragraph could be moved to the results.
Response to comment 10: We aim to briefly summarize our main findings in the first paragraph of the discussion section, and the bottom half of this paragraph summarizes the main takeaways from our subgroup comparisons. We have covered these findings in our results section, but we would be happy to delete the bottom half of the first paragraph in the discussion section if the reviewer thinks necessary.
The evaluation form of reviewer #2:
Response to the evaluation form: We appreciate your evaluation.
